# Identification of Genetic Alterations Associated with Acquired Colistin Resistance in *Klebsiella pneumoniae* Isogenic Strains by Whole-Genome Sequencing

**DOI:** 10.3390/antibiotics9070374

**Published:** 2020-07-02

**Authors:** Myeongjin Choi, Kwan Soo Ko

**Affiliations:** Department of Microbiology, Sungkyunkwan University School of Medicine, Suwon 16419, Korea; vetpharm2@gmail.com

**Keywords:** *Klebsiella pneumoniae*, colistin resistance, CrrAB

## Abstract

The present study was undertaken to find novel genes associated with colistin resistance in *Klebsiella pneumoniae*. Five colistin-resistant mutants were derived from four colistin-susceptible parental *K. pneumoniae* strains belonging to different clones. Whole-genome sequencing was performed for the nine *K. pneumoniae* strains to screen altered candidate genes. Expression levels of genes with amino acid alterations in derivative strains were determined using quantitative real-time Polymerase chain reaction (PCR). Colistin susceptibility was examined in a parental strain complemented with altered candidate genes. Overall, 13 genetic alterations were identified in five pairs of isogenic *K. pneumoniae* strains. Genetic alterations related to *KP1_3468*, including the insertion of an IS*5*-like element in an intergenic or coding region and amino acid substitutions, were identified in three separate derivative strains. Amino acid substitutions and deletion of PhoQ were determined in one derivative strain. With inactivation of CrrA and substituted CrrB, amino acid substitutions and deletion were identified in a repressor of *galETK* operon (*KP1_0061*) and hypothetical protein (*KP1_3620*), respectively. Decreased colistin susceptibility was observed in a parental strain complemented with KP1-0061, but not a KP1-3620 gene. This study demonstrated diverse genetic paths to colistin resistance in *K. pneumoniae*. Our results suggest that a repressor of *galETK* operon may play an important role in colistin resistance in *K. pneumoniae*.

## 1. Introduction

*Klebsiella pneumoniae* is an opportunistic pathogen causing urinary tract infections (UTIs), pneumonia, and bacteremia, particularly in intensive care units. Recently, infections by multidrug-resistant (MDR) or carbapenem-resistant *K. pneumoniae* isolates have become a major concern worldwide [1]. The increase in MDR or carbapenem-resistant *K. pneumoniae* infections has reduced the efficacy of antimicrobial treatments and has raised mortality and morbidity. The limited availability of antimicrobial agents to treat infections caused by Gram-negative pathogens, including *K. pneumoniae,* prompted the resurgence of old antimicrobial agents such as polymyxins (polymyxn B and colistin) [2]. However, the use of colistin induced the emergence of colistin resistance, and even pandrug-resistant isolates, which are resistant to all available antimicrobial agents [3].

Polymyxins are polycationic antimicrobial peptides that interact with the hydrophobic lipid A component of lipopolysaccharides (LPS) in Gram-negative bacteria, which cause bacterial death by disrupting the outer cell membrane [4]. It is known that the modification of LPS following the addition of 4-amino-4-deoxy-l-arabinose to lipid A is related to colistin resistance in *K. pneumoniae* [5]. The modification of LPS is associated with the *pbgPE* operon, which is regulated by two-component regulatory systems, such as PmrAB and PhoPQ [6]. In addition, the eight types of plasmid-mediated *mcr* genes are associated with the addition of phosphoethanolamine to lipid A, which results in colistin resistance [7]. Loss of LPS has been reported in colistin-resistant *Acinetobacter baumannii* isolates [8], but not in *K. pneumoniae*. Recently, insertional inactivation of MgrB, a negative regulator of PhoPQ, was detected in colistin-resistant *K. pneumoniae* isolates [9]. However, mutations in PmrAB, PhoPQ, and MgrB were not found in all colistin-resistant *K. pneumoniae* isolates, and the colistin resistance mechanism has not been fully elucidated. 

Recently, genomic analysis of antimicrobial-susceptible and -resistant bacterial isolates has been used to reveal the mechanisms of antimicrobial resistance [10]. Although several genomic analyses have highlighted colistin resistance mechanisms in *A. baumannii* and *Pseudomonas aeruginosa* [11,12,13], there is a paucity of data regarding the acquired colistin resistance in *K. pneumoniae* [14]. This study focused on identifying genetic alterations associated with colistin resistance among five pairs of *K. pneumoniae* isogenic strains using whole-genome sequencing. 

## 2. Results 

### 2.1. Antimicrobial Susceptibility

Parental and isogenic mutant strains showed the same sequence type (ST) and the same patterns in pulsed-field gel electrophoresis (PFGE) analysis with XbaI (Figure 1). The antimicrobial susceptibilities of the nine *K. pneumoniae* strains are presented in Table 1. While four colistin-susceptible parental strains showed colistin and polymyxin B minimum inhibitory concentrations (MICs) of from 0.5 to 2 mg/L, the colistin and polymyxin B MICs for their isogenic colistin-resistant strains were from >64 to >8196 mg/L. Although the MICs of ciprofloxacin and trimethoprim/sulfamethoxazole were increased in 08-B063R, no significant increase or decrease in the MICs of the other antimicrobial agents was observed in the induced colistin-resistant strains.

### 2.2. Mutations in Colistin-Resistant Isolates

Next-generation sequencing produced 13–17 million nucleotides for each of the nine strains. They covered 90.37–98.37-fold. We were able to map reads from three isolates to 68.61–88.82% and 72.44–81.42% of the *K. pneumoniae* NTUH-K2044 and *K. pneumoniae* MGH 78578 (NC_009648.1) reference genomes, respectively. Overall, the genomic analyses of five pairs of isogenic *K*. *pneumoniae* strains showed 13 nucleotide alterations, including six single nucleotide polymorphisms (SNPs) and seven short insertions and deletions (INDELs). Among the nucleotide alterations, eleven were located in coding sequences, and two were identified in intergenic sequences (IGSs) (Table 2). 

In B0608-134R and B0704-039R1, only IGS alterations were found. In B0608-134R, an IS*5*-like element was inserted between KPN_02065 and KPN_02066. In B0704-039R1, an IS*5*-like element was inserted between KP1_3468 and KP1_3469. 

Alterations in KP1_3468 were also seen in B0704-039R2, a colistin-resistant strain derived from the same parent strain as B0704-039R1. However, these changes were in the coding region. Specifically, three nucleotide substitutions (T26A, C108G, and A128T) led to the amino acid substitutions L9Q, S36R, and K43I, respectively. This strain also contained an IS*5*-like element insertion in *mgrB,* which encodes a PhoPQ regulatory protein.

In B08-063R, four nucleotide alterations were found in three genes, *phoQ*, KP1_3620, and KP1_0061. In *phoQ*, two alterations were found: A803C, leading to a Y268S amino acid substitution, and a 12-bp deletion at the 341st nucleotide position. In KP1_3620, which encodes a hypothetical protein, a 1-bp deletion at the 51st nucleotide position caused a premature stop codon at amino acid position 14. In KP1_0061, which encodes a repressor of the *galETK* operon, a single nucleotide substitution, T104G, caused the amino acid change V35G.

As in B0704-039R2, B0701-068R showed disruptions in MgrB and KP1_3468 caused by IS*5*-like element insertions. In addition, one nucleotide alteration (G593T), leading to the amino acid substitution G198V, was found in KPN_02067, which encodes a RstA regulator (sensor histidine protein kinase).

### 2.3. Orthologs Involved in Colistin Resistance

The KPN_02065, KPN_02066, and KPN_02067 genes were aligned with the *crrAB* region (Locus ID H239_3059-H239_3062) of the UHKPC45 strain. The KPN_02066 genes were uniquely matched against H239_3061 (*crrA*). In addition, KPN_02065 and KPN_02067 genes had a 1-bp mismatch to H239_3062 and 6-bp mismatch to the H239_3060 (*crrB*), respectively. The KP1_3468 gene was matched against the *mgrB* gene (Gene ID 17167876) of the *K. pneumoniae* CG43 stain. The sequence of the KP1_3468 gene is identical to the *mgrB* gene.

### 2.4. Change of mRNA Levels in Candidate Genes Associated with Colistin Resistance

For candidate genes associated with colistin resistance based on genome analysis, mRNA expression levels in colistin-resistant derivatives were compared with those from the susceptible parental strains (Table 2). Genes disrupted by insertion of IS*5*-like elements, such as *mgrB* in B0704-039R2 and B0701-068R, and KP1_3620 in B08-063R, were not included in this analysis. 

For mutations in IGS regions, mRNA expression of the upstream and downstream genes was analyzed to reveal whether mutations in the IGS region could influence nearby gene expression. In B0608-134R, with its IS*5*-like element insertion, KPN_02065 and KPN_02066 expression levels were increased by 208.99- and 4.48-fold respectively, compared with B0608-134. In B0704-039R1, expression of KP1_3468 mRNA was increased 2.51-fold compared with its susceptible parental strain, but KP1_3469 exhibited no significant expression change. 

Expression of KP1_3468 mRNA was increased by 7.91-fold in B0704-039R2 compared with B0704-039. In B08-063R, increased expression was seen in *phoQ* (7.49-fold) and the *galETK* operon repressor, KP1_0061 (2.11-fold), compared to its parental strain. In isolate B0701-068R, there was increased expression of the RstA regulator, KPN_02067 (1.36-fold), but no expression change in KP1_3468 was observed compared with its susceptible parental strain. 

### 2.5. Confirmation of the Role of KP1_0061 and KP1_3620 Gene in Colistin Resistance

To examine whether altered novel candidate genes, KP1_0061 and KP1_3620, from 08-B063R strain were associated with colistin resistance, susceptibility testing to colistin was conducted in the 08-B063 strain harboring a pBAD33 vector containing altered KP1_0061 or KP1_3620 gene. All of the 08-B063 strains harboring a pBAD33 vector induced by 1% L-arabinose showed similar colistin MICs compared to their wild strain, but not 08-B063/pKP1_3620R strain (Table 3). Colistin MIC of 08-B063/pKP1_3620R strain was decreased by 4-fold compared to its parental strain. After the 1 h of incubation with colistin (1 mg/L), 4.7% and 5.3% of bacteria survived in 08-B063 and 08-B063/pBAD33 strains, respectively (Figure 2). The viability of the 08-B063/pKP1_0061R strain against colistin was 2.7-fold higher than its wild-type strain. The viability of the 08-B063/pKP1_3620R strain against colistin could not be determined due to values less than a detection limit of 50 colony-forming unit (CFU)/mL. 

## 3. Discussion

In this study, we used comparative genome analysis to screen for candidate genes associated with colistin resistance in pairs of isogenic strains belonging to four different *K. pneumoniae* clones. ST11 is a dominant clone in Extended-Spectrum β-Lactamase (ESBL)-producing *K. pneumoniae* isolates and is a known cause of UTIs and bacteremia in Korea [15]. ST23 is reported to be a hyper-virulent clone that causes pyogenic liver abscesses with a hyper-mucoviscous phenotype [16]. Although ST152 and ST730 are not seen as frequently as ST11 and ST23, they were used in this study to compare the genetic paths to colistin resistance among *K. pneumoniae* clones. In addition, two colistin-resistant derivatives from one susceptible parental strain were included to determine whether the genetic path to colistin resistance is unique to each strain. 

We identified a total of 13 genetic alterations in five colistin-resistant *K. pneumoniae* mutants, with one to four alterations per strain. Of these, amino acid substitutions in PhoQ and inactivation of MgrB by IS*5*-like elements are well-known mechanisms associated with colistin resistance in *K. pneumoniae* [5]. In this study, PhoQ variations and MgrB inactivation were identified in only one and two colistin-resistant derivatives, respectively. No PhoQ variation or MgrB inactivation was identified in two of the mutants (B0608-134R and B0704-039R1). Instead, these mutants had genetic alterations in intergenic regions. 

In B0608-134R, an IS*5*-like element was inserted into the intergenic region between KPN_02065 and KPN_02066. These two genes, which encode a putative transport protein and response regulator, were highly expressed in a colistin-resistant mutant. This suggests that the insertion of an IS*5*-like element in the IGS may affect the expression and/or function of these two proteins. KPN_02066 encodes a response regulator harboring a CheY-like receiver domain. Response regulators with a CheY-like receiver domain are known to be associated with signal transduction and transcription in *Deinococcus radiodurans* [17]. Although the precise mechanism is unknown, the modification of signal transduction and transcription may be associated with colistin resistance in this strain. 

One of the interesting findings in this study is that genetic alterations related to KP1_3468 occurred repeatedly in either the nearby IGS region (B0704-039R1) or as amino acid substitutions in the coding sequence (B0704-039R2). Although B0704-039R1 contained only a single IGS insertion between KP1_3468 and KP1_3469, KP1_3468 alterations in B0704-039R2 and B0701-068R were accompanied by disruptions such as MgrB inactivation. Thus, the association of KP1_3468 with colistin resistance is not clear, and further investigation is required. It is also notable that two colistin-resistant derivatives from one susceptible parental strain showed different genetic variations in MgrB. Colistin-resistant mutants derived from the same susceptible strain showed different amino acid substitutions reported in *A. baumannii* [18]. 

The repeated finding of IS*5*-like element insertions in colistin-resistant derivatives is also noteworthy. As an IS*5*-like element is already present in the genome of susceptible parental strains (data not shown), it is likely to originate endogenously [5]. It is thought that IS elements play beneficial roles in adaptive evolution in bacteria by generating genetic diversity [19]. Our results suggest that mobile IS*5*-like elements can facilitate the development of colistin resistance after colistin exposure in *K. pneumoniae*. 

In B08-063R, amino acid substitution and upregulation of KP1_0061, which encodes a *galETK* operon repressor, were identified in addition to PhoQ alterations. Although there were no changes in MICs between a parental strain with or without complementation with altered KP1_0061, decreased susceptibility in the presence of 1 mg/L colistin was observed in complementation strains. The *galETK* operon is associated with galactose metabolism, and the product Uridine diphosphate (UDP)-galactose contributes to the early stages of LPS biosynthesis in *K. Pneumoniae* [20]. Upregulation of KP1_0061 may repress the *galETK* operon, leading to obstruction of LPS synthesis. It is speculated that obstruction of LPS synthesis due to repression of the *galETK* operon might decrease the interaction between colistin and the outer membrane. Unfortunately, one of the candidate genes, KP1_3620, found through next-generation sequencing failed to reduce susceptibility to colistin. Further studies are needed to determine whether these genes were associated with decreased susceptibility to trimethoprim-sulfamethoxazole in 08-B063R in this study. 

The discovery of an amino acid substitution in a RstA regulator in B0701-068R was also interesting because the RstA operon is activated by the PhoPQ system in low Mg^2+^. It is also associated with virulence factors such as biofilm formation or iron transport in Negative bacteria [21,22]. It is hypothesized that overexpression of the PhoPQ system due to the inactivation of MgrB may result in colistin resistance, while downstream, the RstAB system may lead to increased fitness of a colistin-resistant strain.

In this study, genetic alterations in colistin-resistant derivatives of *K. pneumoniae* were investigated by whole-genome sequencing. Five pairs of isogenic colistin-susceptible and -resistant strains did not show identical genetic alterations, implying that there are diverse genetic paths to colistin resistance. In addition to well-known *phoQ*, *mgrB,* and *crrAB* associated with colistin resistance, other candidate genes such as KP1_3468, and KP1_0061 (encoding a *galETK* operon repressor) were found to be associated with colistin resistance in *K. pneumoniae*. We also showed that an IS*5*-like element might play a role in the evolution of antimicrobial resistance. Further study is required to investigate the roles of these genes and IS in the emergence of colistin resistance.

## 4. Materials and Methods

### 4.1. Bacterial Strains

Four colistin-susceptible *K. pneumoniae* strains (B0608-134, B0704-039, 08-B063, and B0701-068) isolated from patients during 2006–2007 in Korea were used in this study. To determine colistin resistance mechanisms in diverse lineages of *K. pneumoniae*, four strains belonging to different clones (ST730, ST11, ST23, and ST152) according to multi-locus sequence typing (MLST) analysis were used (Table 1) [23]. To induce resistance to colistin, the strains were repeatedly cultured with increasing concentrations of colistin in lysogeny broth (LB). The colistin-resistant mutants (B0608-134R, B0704-039R1, B0704-039R2, 08-B063R, and B0701-068R) were selected in LB containing colistin ranging from 2 to 64 mg/L. B0704-039R1 and B0704-039R2 were derived from the same susceptible strain in different flasks. 

### 4.2. Pulsed-Field Gel Electrophoresis (PFGE)

To investigate the clonal relationship between parental and mutant strains, PFGE was carried out as previously described [24]. Briefly, Genomic DNA was digested with XbalI (New England Biolabs, Beverly, MA, USA) and the fragments were separated in a 1% agarose gel using CHEF DR II-apparatus (Biorad, Hercules, CA, USA). The electrophoresis conditions were a constant voltage of 6 V/cm with pulse times from 5 to 90 s for 18 h. Lambda ladders (New England Biolabs, Beverly, MA, USA) were applied as molecular size markers. PFGE gels were stained with Ethidium bromide, visualized by ultraviolet light (BioRad, Berkeley, CA, USA) and converted to TIFF files.

### 4.3. In Vitro Antimicrobial Susceptibility Testing

Minimum inhibitory concentrations (MICs) of antimicrobial agents were determined by Clinical and Laboratory Standards Institute (CLSI) broth microdilution methods [25]. Eleven antimicrobial agents were tested: colistin, polymyxin B, ampicillin, aztreonam, cefotaxime, ceftazidime, ciprofloxacin, gentamicin, imipenem, piperacillin/tazobactam, and trimethoprim/sulfamethoxazole. The results were interpreted based on susceptibility interpretive criteria established in the CLSI standard M100-S28 [25]. Quality control was performed using *E. coli* ATCC 25922, and *Pseudomonas aeruginosa* ATCC 27853 in three independent experiments. *K. pneumoniae* isolates would be considered susceptible to colistin and polymyxin B at breakpoints of ≤2 mg/L [26,27].

### 4.4. Genome Sequencing and Confirmation of Genetic Variants

Genomic DNA was extracted from overnight cultures of nine *K. pneumoniae* strains by using a G-spin^TM^ Genomic DNA extraction Kit (iNtRON Biotechnology, Inc., South Korea). Purified DNA with a 260/280 purity ratio of 1.75–2.0 was sequenced using Illumina Hiseq2000 Preliminary Performance Parameters, with 100X coverage (Macrogen; Seoul, South Korea). After sequencing, raw sequence data were filtered based on quality score and then assembled onto the *K. pneumoniae* NTUH-K2044 (NC_012731.1) and *K. pneumoniae* MGH 78578 (NC_009648.1) reference genomic sequences using Burrows-Wheeler Aligner v0.5.9-r16. Sequence features, including SNPs and INDELs, in colistin-resistant strains were identified through next-generation sequencing and confirmed through Sanger sequencing, using the primers listed in Table 4. 

### 4.5. Identification of Orthologous Genes 

To exclude genes known to be the cause of colistin resistance in *K. pneumoniae* from complementation analysis, KPN_02065, KPN_02066, KPN-02067, and KP1_3468 genes were matched against the *crrAB* region and *mgrB* gene [5,28]. The *crrAB* region and *mgrB* sequence database was provided by the National Center for Biotechnology Information (NCBI). A comparison of the orthologous sequence was analyzed by DNAstar EditSeq and MegAline (DNASTAR, Inc., Madison, WI, USA).

### 4.6. mRNA Expression Analysis

Based on the whole-genome sequencing data, the expression levels of candidate genes associated with colistin resistance were investigated using quantitative real-time PCR in five pairs of isogenic *K. pneumoniae* strains, as previously described [29]. Briefly, total RNA was used as the template for reverse transcription according to the manufacturer’s protocol of the Omniscript reverse transcriptase kit (Qiagen, Hilden, Germany). SYBR green PCR master mix (Applied Biosystems, CA, USA) was used for the amplifications in a Thermal Cycler Dice Real-Time System TP800 (Takara Bio Inc., Shiga, Japan). The thermal cycling conditions included an initial denaturation step at 95 °C for 10 min, followed by 30 cycles at 95 °C for 30 s, 60 °C for 30 s, and 72 °C for 30 s. The fold changes in expression in mutant strains compared to parental strains were determined using the primers listed in Table 4. Gene expression was analyzed as a relative mRNA level normalized to a housekeeping gene, *rpoB*. Three independent experiments were performed for analysis. 

### 4.7. Complementation Assay

To determine whether the mutation in novel candidate genes was necessary to confer colistin resistance, the pBAD33-derived plasmids were purified from *Escherichia coli* DH5α (American Type Culture Collection, VA, USA). The plasmid vectors carrying the altered *KP1_0061* or *KP1_3620* gene were introduced into the 08-B063 strain. To construct the pKP1_0061R vector, PCR-generated fragments harboring the *KP1_0061* coding region of the 08-B063R strain were cloned into the SacI and HindIII (New England Biolabs, MA, USA) sites of multiple cloning site (MCS) in the pBAD33 vector. To construct the pKP1_3620R vector, PCR-generated fragments harboring the *KP1_3620* coding region of the 08-B063R strain were cloned into the SmaI and PstI site of multiple cloning sites (MCS) in pBAD33 vectors. These vectors were used to transform *E. coli* DH5α cells. The vectors were purified from the transformants and confirmed using restriction analysis with enzyme treatments and Sanger sequencing. Electroporation of vector into the 08-B063 strain was conducted as described [30]. The transformed cells were selected on an agar plate containing 100 mg/L chloramphenicol. The presence of these plasmids in the colonies was confirmed using restriction analysis and PCR, with the primers pBAD-MCS-F and pBAD-MCS-R. The strains each carrying pBAD33, pKP1_0061R, or pKP1_3620R were grown in the presence of both chloramphenicol 25 mg/L and 1% L-arabinose.

### 4.8. Colistin Susceptibility Testing

Colistin susceptibility testing was performed as described with some modifications [6]. In brief, the overnight-grown 08-B063 strain carrying the pBAD33, pKP1_0061R, or pKP1_3620R vector was obtained from Mueller Hinton Broth (MHB) in the presence of both chloramphenicol 25 mg/L and 1% L-arabinose. The bacteria were then washed twice and a suspension containing ca. 1.5 × 10^6^ CFU/mL in MHB was prepared. Then, 100 μL of the suspension was placed in each well of a 96-well micro-titer plate and 100 μL MHB or MHB-diluted colistin was added to each well to final concentrations of 1 mg/L of colistin. The plate was incubated at 37 °C for 1 h. Subsequently, 100 μL of the suspension was directly plated on Mueller Hinton agar plates and incubated at 37 °C overnight to determine the number of viable bacteria. The survival rates were expressed as colony counts divided by the number of the same culture treated with MHB and multiplied by 100. The results were shown as the average ± standard deviation. Changes in the colistin susceptibility of mutants complemented with pKP1_0061R or pKP1_3620R vectors were examined using the quantitative bactericidal assay.

## 5. Conclusions

In all mutants derived from different clones, genetic alterations were observed in well-known *phoQ*, *mgrB*, and *crrAB* genes associated with acquired colistin resistance. In addition, an intrinsic IS5-like element was found in intergenic or intragenic regions of those genes. However, two novel candidate genes such as KP1_0061 and KP1_3620 were discovered in one colistin-resistant mutant. Together with the divergent alterations between two mutants originated from the same parental strain, these results indicate diverse genetic paths to colistin resistance in *K. pneumoniae*. This study will provide a better understanding of bacterial evolution against colistin resistance in *K. pneumoniae*. 

## Figures and Tables

**Figure 1 antibiotics-09-00374-f001:**
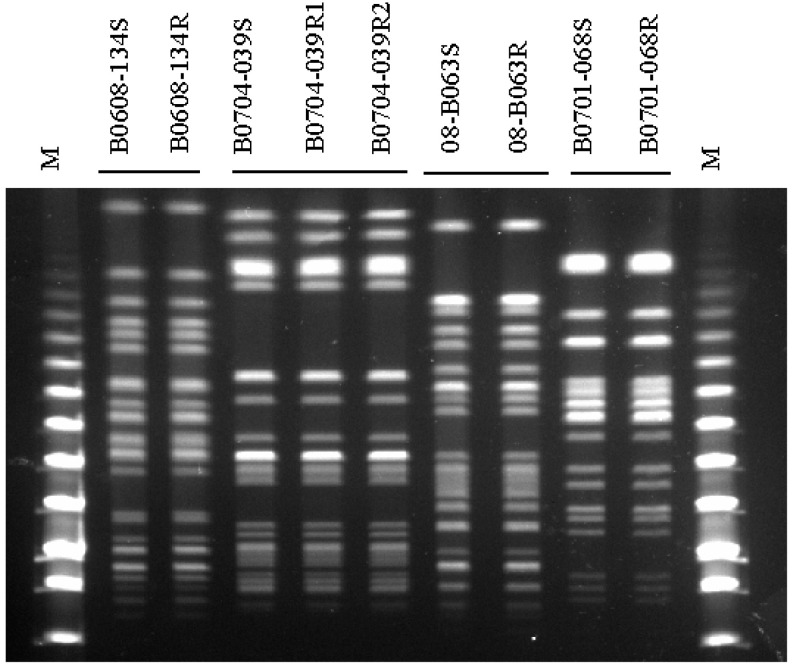
Pulsed-field gel electrophoresis (PFGE) profiles obtained with restriction enzyme XbaI.

**Figure 2 antibiotics-09-00374-f002:**
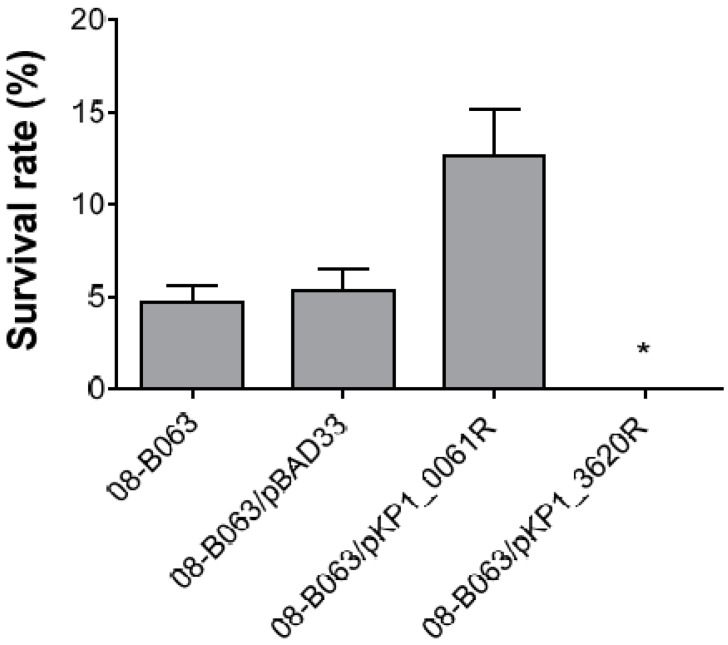
Colistin susceptibility testing in the 08-B063 strain carrying the pKP1_0061R or pKP1_3620R vector. The strains were challenged with 1 mg/L of colistin for 1 h. * After 1 h incubation with colistin, survival of 08-B063/pKP1_3620R could not be detected due to values less than a detection limit of 50 CFU/mL.

**Table 1 antibiotics-09-00374-t001:** Antimicrobial susceptibility profiles in four pairs of isogenic *Klebsiella pneumoniae* strains.

Isolate	ST ^a^	Minimum Inhibitory Concentration(mg/L) ^b^										
		COL	PB	AMP	AZ	CTX	CAZ	CIP	GEN	IMI	P/T	SXT
B0608-134	730	1	2	64	≤0.06	≤0.06	1	≤0.06	1	0.125	16/4	0.06/1.18
B0608-134R		256	>64	64	≤0.06	≤0.06	1	≤0.06	2	0.25	16/4	0.06/1.18
B0704-039	11	0.5	2	>64	>64	>64	>64	>64	>64	1	>64/4	4/76
B0704-039R1		>8192	>64	>64	>64	>64	>64	>64	>64	1	>64/4	4/76
B0704-039R2		>128	>64	>64	>64	>64	>64	>64	>64	0.5	>64/4	4/76
08-B063	23	0.5	2	64	≤0.06	≤0.06	0.5	≤0.06	0.5	0.125	8/4	0.12/2.37
08-B063R		512	>64	>64	≤0.06	≤0.06	0.5	0.125	0.5	0.125	16/4	0.5/9.5
B0701-068	152	0.5	2	64	≤0.06	≤0.06	1	≤0.06	1	0.25	16/4	0.06/1.18
B0701-068R		1024	>64	>64	≤0.06	≤0.06	1	≤0.06	1	0.25	16/4	0.125/2.37

^a^ ST, Sequence type in multi-locus sequence typing. ^b^ COL, colistin; PB, polymyxin B; AMP, ampicillin; AZ, aztreonam; CTX, cefotaxime; CAZ, ceftazidime; CIP, ciprofloxacin; GEN, gentamicin; IMI, imipenem; P/T, piperacillin/tazobactam; SXT, trimethoprim/sulfamethoxazole.

**Table 2 antibiotics-09-00374-t002:** Nucleotide and amino acid alterations and mRNA expression changes among five pairs of isogenic *Klebsiella pneumoniae* strains.

Set	Gene Name	Function	Nucleotide Alteration	Amino Acid Alteration	Fold Change of mRNA Expression
B0608-134 versus B0608-134R	Intergenic region between KPN_02065 and KPN_02066	KPN_02065, putative transport protein KPN_02066, putative response regulator consisting of a CheY-like receiver domain and a winged-helix DNA-binding domain	Insertion of Insertion sequence (IS)*5*-like element	-	208.99 ± 48.27-fold increase for KPN_02065
					4.48 ± 0.1-fold increase for KPN_02066
B0704-039 versus B0704-039R1	Intergenic region between KP1_3468 and KP1_3469	KP1_3468; conserved hypothetical protein KP1_3469; putative outer membrane	Insertion of IS*5*-like element	-	2.51 ± 0.49-fold increase for KP1_3468
					1.03 ± 0.11 fold increase for KP1_3469
B0704-039 versus B0704-039R2	KP1_3468	Hypothetical protein	T26A, C108G, A128T	L9Q, S36R, K43I	7.91 ± 0.31-fold increase
	*mgrB*	PhoPQ regulatory protein	Insertion of IS*5*-like element	Frameshift	-
B08-063 versus B08-063R	*phoQ*	Sensor protein PhoQ	A803C	Y268S	7.49 ± 0.52-fold increase
			12-bp deletion at 341st nucleotide	Deletion of 5 amino acids at position 114	
	KP1_3620	Hypothetical protein	1 bp deletion at 51st nucleotide	Premature stop at position 14	-
	KP1_0061	Repressor of *galETK* operon	T104G	V35G	2.11 ± 0.35-fold increase
B0701-068 versus B0701-068R	KP1_3468	Hypothetical protein	Insertion of IS*5*-like element	Frameshift	0.9 ± 0.14-fold increase
	KPN_02067	Sensor histidine protein kinase (RstA regulator)	G593T	G198V	1.36 ± 0.05-fold increase
	*mgrB*	PhoPQ regulatory protein	Insertion of IS*5*-like element	Frameshift	-

**Table 3 antibiotics-09-00374-t003:** Colistin MICs in 08-B063 strain harboring a pBAD33 vector containing altered genes.

Isolates	Colistin MICs(mg/L)
08-B063	0.5
08-B063/pBAD33	0.5
08-B063/pKP1_0061R	0.5
08-B063/pKP1_3620R	0.125

**Table 4 antibiotics-09-00374-t004:** Primers used in this study.

Primer	Sequence (5′→ 3′)	Amplicon Size (bp)	Reference
Primers for sequencing			
134-target1-F	TGAATGCACAAGGTAAAGCCAGG	500	This study
134-target1-R	CCGTCTTCGGCTCTTAAGGTTTT		
039-target1-F	GACACAGCCAGCGATGCCAG	416	This study
039-target1-R	TTGTATGATCCATGGCGTGA		
063-target1,2-F	ATGCAGGTAATAACCAATCT	751	This study
063-target1,2-R	ATCCCGGAAAACCTGAATAT		
063-target3-F	CCATAATAAAAAAAATAATC	362	This study
063-target3-R	GCCCCTCCCCACACATCTTT		
063-target4-F	ACGGCCGAGCGCCAGAAACG	720	This study
063-target4-R	TCGCATCCGCATCCGCCAGG		
068-target1-F	TATGATGCACACCTGTCGGG	780	This study
068-target1-R	CACTGTGGAATAACACCCCA		
068-target2-F	GGCACTGAATAATGCGTAAG	480	This study
068-target2-R	GCCTGAAACAAATCTCTCAG		
mgrB-ext-F	CAGCCAGCGATGCCAGATTT	380	[5]
mgrB-ext-R	CCTGGCGTGATTTTGACACGA		
Primers for qRT-PCR			
RT-KPN_02065-F	CCCCGGGGTGCTATCTCAC	135	This study
RT-KPN_02065-R	CCGCCCAGGATGGCATTTAT		
RT-KPN_02066-F	TCGCTGCCATACTGACAGGCT	148	This study
RT-KPN_02066-R	TTCCAGCCATCGTACACGGG		
RT-KP1_3468-F	AAAAATTACGGTGGGTTTTACTGA	109	This study
RT-KP1_3468-R	ATGCCGCTGAAAAACTGAAC		
RT-KP1_3469-F	TGGATTGGCGTATTATTGAGC	122	This study
RT-KP1_3469-R	TGAACATAAAGTGCGGTGCT		
RT-phoQ-F	TGCCAGGGAAGCGGACTAC	100	This study
RT-phoQ-R	GCGGCGGATCAGTGATAAAC		
RT-KP1_0061-F	TTATCACCGACTGCAACACC	138	This study
RT-KP1_0061-R	CGACATCGGTCTCAATGATG		
RT-KPN-02067-F	GGGAAACCTTAGTGCCAGAGC	111	This study
RT-KPN-02067-R	TGCGCATCCAGCGTCTGTAA		
RT-rpoB-F	CGCGTATGTCCGATCGAAA	100	This study
RT-rpoB-F	GCGTCTCAAGGAAGCCATATTC		

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
