# Peer review of "Identification of Genetic Alterations Associated with Acquired Colistin Resistance in Klebsiella pneumoniae Isogenic Strains by Whole-Genome Sequencing"

_antibiotics, 2020, doi:10.3390/antibiotics9070374_

Round 1

Reviewer 1 Report

Review of the article: “Identification of genetic alterations associated with acquired colistin resistance in Klebsiella pneumoniae isogenic strains by whole-genome sequencing”

Manuscript ID: antibiotics-841313

I have read the manuscript with a great intererst. In my opinon all experiments were well planned and performer. The obtained results are interesting and important for understanding and explanation of mechanisms K. pneumoniae resistance against colistin. I can reccomend the  mansucript for publication in Antibiotics. However, I have some critical remarks (minor of importance) about current version of the article, which should be taken into account before final acceptance. Detailed comments are presented below.

Abstract – well prepared, no critical remarks

Introduction – I think that the authors should add some more basic information about polymyxins – not all potential readers are familiar with this group of antimicrobial agents. Lines 44-45, the bracket is open but it is not closed.

Results – In my opinion some important results are presented in the section Materials and methods – point 4.1.. These results should be moved to Results (including the Figure 1).

Page 5 line 81 – the abbreviations SNPs and INDELs should be explained (they are explained but I found it on the page 10 – it should be done earlier on the page 5)

Materials and methods – Section 4.5 there is no information which method was used for calculation of fold changes in expression of the selected genes in parental strains and mutants. The authors have not given the borderline value of fold of change of expression level of these genes which should be considered as important from the point of view of resistance.

The authors have not presented any details about conditions of RT-PCR reactions (except of sequences of primers).

Discussion – well presented, no critical comments.

Final decision – acceptance after minor revision.

Author Response

Editor-in-Chief

Antibiotics

June 25, 2020

Attached for your consideration for publication in “Antibiotics” is the revised manuscript entitled “Identification of genetic alterations associated with acquired colistin resistance in Klebsiella pneumoniae isogenic strains by whole-genome sequencing by Drs Choi M and Ko KS.

We appreciate an Editor and three reviewers for critical review of our manuscript. We tried to revise all points mentioned by reviewers. Revised portions are highlighted.

We attached the reply to Reviewer’s opinion.

If you require any additional information, please let us know. We look forward to receiving your decision regarding publication.

Reviewer 2 Report

The study is well designed, and data analysis is rather extensive and in-depth. It can be accepted without any further changes.

Author Response

(The authors gave the same response as above.)

Reviewer 3 Report

In the manuscript „Identification of genetic alterations associated with acquired colistin resistance in Klebsiella pneumoniae 3 isogenic strains by whole-genome sequencing” Myeongjin Choi and Kwan Soo Ko, the authors reported on the genetic basis of colistin resistance in Klebsiella pneumoniae. Therefore, colistin-resistant Klebsiella mutants from different clonal lineages were subjected to WGS for bioinformatics analysis. Based on the sequencing data candidate genes with altered sequences were subjected to qPCR to determine their expression levels. Furthermore, colistin susceptibility was tested in complementation studies. The authors had found alterations in KP1_3468, PhoQ, CrrA, CrrB, galETK operon (KP1_0061) and KP1_3620 of which some of them were confirmed to be highly implicated in colistin resistance development.

This reviewer feel that the manuscript presents very interesting information on colistin resistances. However, this reviewer misses also an short introduction of mobile colistin resistance as they are of higher impact for colistin resistance development. The manuscript s well written and the results are well described. Further attention in the revision should be focused on the materials and methods section. Some further information to the experimental procedures are needed. Furthermore, this reviewer feels that companies and locations are not given for all materials, vectors used.

Additionally, some minor spelling and typos errors need to be revised.

Introduction: Please also include a statement on the presence of mobile colistin resistances and the number, diversity and impact of the genes for colistin resistance dissemination.

Table 1: How can the increased resistance of for 08-B063 wildtype and mutant isolate for SXT be explained… how often were the experiments conducted

Figure 1 should be shifted to the results section

Line 216: LB medium is not Luria Bertani it is lysogeny broth as written in a review of Bertani!

Line 220: PFGE analysis: please specify the protocol and the conditions used

Line 223: which quality control isolate has been used as internal control

Line 232: how was the DNA prepared (quality parameters!)

Line 251: what does it means three independent experiments or one experiment with three technical replicates… please specify

Author Response

(The authors gave the same response as above.)

Round 2

Reviewer 3 Report

The requested revision were conducted appropriately.